# Physiological and Biochemical Regulation Mechanism of Exogenous Hydrogen Peroxide in Alleviating NaCl Stress Toxicity in Tartary Buckwheat (*Fagopyrum tataricum* (L.) Gaertn)

**DOI:** 10.3390/ijms231810698

**Published:** 2022-09-14

**Authors:** Xin Yao, Meiliang Zhou, Jingjun Ruan, Yan Peng, Chao Ma, Weijiao Wu, Anjing Gao, Wenfeng Weng, Jianping Cheng

**Affiliations:** 1College of Agronomy, Guizhou University, Guiyang 550025, China; 2Institute of Crop Science, Chinese Academy of Agriculture Science, Beijing 100081, China

**Keywords:** Tartary buckwheat, H_2_O_2_, NaCl, physiology and biochemistry

## Abstract

We aimed to elucidate the physiological and biochemical mechanism by which exogenous hydrogen peroxide (H_2_O_2_) alleviates salt stress toxicity in Tartary buckwheat (*Fagopyrum tataricum* (L.) Gaertn). Tartary buckwheat “Chuanqiao-2” under 150 mmol·L^−1^ salt (NaCl) stress was treated with 5 or 10 mmol·L^−1^ H_2_O_2_, and seedling growth, physiology and biochemistry, and related gene expression were studied. Treatment with 5 mmol·L^−1^ H_2_O_2_ significantly increased plant height (PH), fresh and dry weights of shoots (SFWs/SDWs) and roots (RFWs/RDWs), leaf length (LL) and area (LA), and relative water content (LRWC); increased chlorophyll a (Chl a) and b (Chl b) contents; improved fluorescence parameters; enhanced antioxidant enzyme activity and content; and reduced malondialdehyde (MDA) content. Expressions of all stress-related and enzyme-related genes were up-regulated. The *F3′H* gene (flavonoid synthesis pathway) exhibited similar up-regulation under 10 mmol·L^−1^ H_2_O_2_ treatment. Correlation and principal component analyses showed that 5 mmol·L^−1^ H_2_O_2_ could significantly alleviate the toxic effect of salt stress on Tartary buckwheat. Our results show that exogenous 5 mmol·L^−1^ H_2_O_2_ can alleviate the inhibitory or toxic effects of 150 mmol·L^−1^ NaCl stress on Tartary buckwheat by promoting growth, enhancing photosynthesis, improving enzymatic reactions, reducing membrane lipid peroxidation, and inducing the expression of related genes.

## 1. Introduction

Buckwheat (*Fagopyrum esculentum* Moench, 2n = 8) is an annual or perennial grain crop used as food and forage [1]. In China, buckwheat is divided into three main cultivars: Tartary buckwheat (*F. tataricum*), sweet buckwheat (*F. esculentum*), and golden buckwheat (*F. cymosum*), in addition to wild species [2]. Tartary buckwheat is a dicotyledonous plant in the family Polygonaceae [3]. China is not only the largest producer of Tartary buckwheat, but also the global center of buckwheat diversity. Tartary buckwheat is suitable for growing in high-altitude southwestern regions of China, such as Yunnan, Guizhou, and Sichuan Provinces, which have a cold climate and short frost-free period [4]. The southwest region of China is internationally recognized as the center of origin for buckwheat [5]. China ranks among the highest in the world in terms of planting area and output, with a current output second only to Russia. Tartary buckwheat has been promoted, grown, and eaten in many countries in the world [6]. Tartary buckwheat is rich in flavonoids and other biologically active substances, including rutin, quercetin, kaempferol, morin, and other natural compounds, and the rutin is the main essential ingredient in Tartary buckwheat. The content of rutin in the shoots is significantly higher than that in the roots [7]. Tartary buckwheat also contains a variety of nutrients, such as protein, starch, and vitamins, which can relieve cardiovascular sclerosis, diabetes, and other diseases [4,8]. Therefore, with the rising demand for Tartary buckwheat products, its development and utilization have gained increasing attention.

In natural biological populations, interactions between genotype and environment affect the phenotype of plants. In plants with the same genotype, different environments can lead to various phenotypic differences [9,10]. With increasingly serious global warming and rising land evaporation, the problem of soil salinization has become prominent in countries at middle and low latitudes [11]. Salt stress refers to the adverse effect of excess soil salt on plants [12], and is a major abiotic stress that seriously affects the growth and development of plants, inhibits their physiological and biochemical metabolic processes, even causing plant death, greatly reduces agricultural yields, and restricts agricultural development potential [13,14]. When plants are subjected to salt stress, with increasing salt concentration, the growth trend of plants decreases [15], the content of reactive oxygen species (ROS) increases [16], the activity of antioxidant enzymes reduces [17], photosynthesis weakens [18], and the homeostasis of sodium (Na^+^) and potassium (K^+^) ions is unbalanced [19]. Therefore, the improvement, development, and utilization of soil salinization is an effective way to improve the growth environment of plants, and improving plant growth and yield under salt stress has become a hot research topic globally.

Reactive oxygen species (ROS) are biochemical substances that play a vital role in seed dormancy and germination. In particular, the intracellular homeostasis of hydrogen peroxide (H_2_O_2_), superoxide anion (O_2_^−^), and hydroxyl radical (OH^−^) is involved in signaling cascades, and this determines the growth and development process and stress response [20]. H_2_O_2_ is an important regulatory component of plant signal transduction. It is not only a free radical produced by oxidative stress products, but also helps to maintain plant cell homeostasis [21]. Concurrently, H_2_O_2_ can regulate the expression of various genes in physiological metabolism, including the genes encoding antioxidant enzymes, regulating biotic and abiotic stress response proteins [22,23]. H_2_O_2_ has a concentration-dependent effect on physiological and biochemical processes. That is, high concentrations of H_2_O_2_ produce an oxidative stress reaction in plants, leading to cell damage [24]; whereas at low concentrations, H_2_O_2_ acts as a signal molecule participating in tolerance to various abiotic [25] and biological stresses [26], and also plays a regulatory role. A number of studies have reported that H_2_O_2_ is involved in the regulation of physiological activities such as seed germination [27] and photosynthesis [28].

As a multifunctional signaling molecule, H_2_O_2_ plays an important role in a series of physiological and biochemical processes. However, the involvement of H_2_O_2_ in the anti-stress metabolism of Tartary buckwheat, and particularly the mechanism of salt tolerance, is rarely studied. In previous research by the research group, it was found that 1–5 mmol·L^−1^ H_2_O_2_ (without NaCl treatment) could significantly increase the root growth of Tartary buckwheat at the germination stage [29], and at the germination stage 5–10 mmol·L^−1^, H_2_O_2_ can significantly promote the germination of Tartary buckwheat under 50 mmol·L^−1^ NaCl treatment [1]. Therefore, in this study, different concentrations of exogenous H_2_O_2_ were applied to the leaves of Tartary buckwheat to explore the growth, photosynthesis, antioxidant enzyme activity, membrane lipid peroxidation, and expression of related genes of Tartary buckwheat under NaCl stress. The physiological metabolic mechanism by which H_2_O_2_ regulates the salt tolerance of Tartary buckwheat was evaluated through correlation and principal component analyses to provide a scientific theoretical basis for exogenous H_2_O_2_ treatment to improve the salt tolerance of Tartary buckwheat.

## 2. Results

### 2.1. Effect of Exogenous H_2_O_2_ on Plant Height and Biomass under NaCl Stress

Figure 1 shows that the growth of Tartary buckwheat under the CK treatment (H_2_O + H_2_O) and 5H + N (5 mmol·L^−1^ H_2_O_2_ + 150 mmol·L^−1^ NaCl) treatments was significantly better than that under NaCl (150 mmol·L^−1^ NaCl + H_2_O) and 10H + N (10 mmol·L^−1^ H_2_O_2_ + 150 mmol·L^−1^ NaCl) treatments, and the root growth with 5H + N was better than that with the other three treatments.

Under NaCl stress, plant height (PH), stem fresh weight (SFW), stem dry weight (SDW), root fresh weight (RFW), and root dry weight (RDW) of Tartary buckwheat were significantly decreased by 28.22%, 51.90%, 44.63%, 59.60%, and 48.19%, respectively, compared with what they were under the CK treatment (Figure 2A–E). Under the 10H + N treatment, PH, SFW, SDW, RFW, and RDW were significantly decreased by 0.78-fold, 0.43-fold, 0.63-fold, 0.58-fold, and 0.61-fold, respectively, relative to what they were under the CK treatment (Figure 2A–E). The SFW of Tartary buckwheat decreased by 9.80% under the 10H + N treatment compared with what it was under the NaCl treatment (Figure 2B). Under the 5H + N treatment, the PH, SFW, SDW, RFW, and RDW of Tartary buckwheat were significantly higher than they were under the NaCl treatment (Figure 2A–E); SFW and RFW increased by 124.60% and 141.58%, respectively (Figure 2B,D), and SFW was significantly decreased by 2.49 times under 10H + N treatment (Figure 2B).

### 2.2. Effect of Exogenous H_2_O_2_ and NaCl Stress on Leaf Growth and Relative Water Content

Compared with the CK, the leaf area (LA) and leaf length (LL) of Tartary buckwheat increased significantly by 24.75% and 18.60%, respectively, under 5H + N treatment (Figure 3A,B), and the leaf relative water content (LRWC) decreased slightly (Figure 3C); under 10H + N treatment, LA was significantly decreased by 11.88% (Figure 3A), and LL and LRWC were significantly decreased by 23.97% and 22.84%, respectively (Figure 3B,C). Compared with NaCl treatment, the LA, LL, and LRWC of Tartary buckwheat were significantly increased under the 5H + N treatment (Figure 3A–C), among which LL increased the most (91.33%) (Figure 3C). LA, LL, and LRWC were significantly increased under 10H + N treatment; however, the increasing trend of LRWC was not significant, and all values were significantly lower under 10H + N treatment than under 5H + N treatment (Figure 3A–C).

### 2.3. Effect of Exogenous H_2_O_2_ and NaCl Stress on Leaf Photosynthetic Pigments

Compared with CK, Chl a, Chl b, and Car were significantly reduced by 40.63–69.33% and 31.69–55.11%, respectively, under the NaCl stress and 10H + N treatments (Figure 4A). Under 5H + N treatment, Chl a, Chl b, and Car showed an insignificant downward trend (Figure 4A). Compared with NaCl stress, Chl a, Chl b, and Car were significantly increased in Tartary buckwheat under 5H + N treatment (Figure 4A), among which Car showed the largest increase (191.30%) (Figure 4A). Under 10H + N treatment, Chl a was significantly decreased, while Chl b and Car were significantly increased (Figure 4A).

The total chlorophyll content included Chl a and Chl b. As shown in Figure 4B,C, the maximum observed value of Chl a + b was in the CK treatment (1.147), and the maximum Chl a/b was in the NaCl treatment (2.639). Under the 10H + N treatment, Chl a + b and Chl a/b of Tartary buckwheat were significantly decreased compared with what they were under CK treatment (Figure 4B,C). Under NaCl stress, Chl a + b was significantly decreased (Figure 4B), while Chl a/b showed no significant increase compared to under CK treatment (Figure 4C). Under 5H + N treatment, Chl a + b of Tartary buckwheat was significantly increased (56.61%) compared with under NaCl stress (Figure 4B).

### 2.4. Effect of Exogenous H_2_O_2_ and NaCl Stress on Leaf Fluorescence Parameters

The change trends of PSII maximum photochemical efficiency (Fv/Fm), effective quantum yield (Fv’/Fm’), non-photochemical quenching coefficient (NPQ), and effective electron transfer rate (ETR) of Tartary buckwheat were similar (Figure 5A,B,E,F). Compared with CK, the Fv/Fm, Fv’/Fm’, NPQ, and ETR of Tartary buckwheat under NaCl stress were significantly decreased by 25.74%, 27.69%, 30.86%, and 36.26%, respectively (Figure 5A,B,E,F). The Fv/Fm, Fv’/Fm’, NPQ, and ETR of Tartary buckwheat under 10H + N treatment were all reduced (Figure 5A,E,F), with the largest reduction observed in NPQ (29.14%) (Figure 5E); this reduction was significant for all except Fv’/Fm’. Compared with NaCl stress, the Fv/Fm, Fv’/Fm’, NPQ, and ETR were significantly increased by 22.70–82.66% (Figure 5A,B,E,F), with the largest increase observed in the ETR (Figure 5F).

The change trends of Φ_PSII_ and qP of Tartary buckwheat were similar (Figure 5C,D). Compared with CK, Φ_PSII_ and qP were significantly increased under NaCl stress and the 10H + N treatment. The largest increases in Φ_PSII_ and qP (1.36 and 1.45 times, respectively) were observed under 10H + N treatment (Figure 5C,D), while Φ_PSII_ and qP did not significantly change with 5H + N treatment (Figure 5C,D). Compared with 5H + N treatment, Φ_PSII_ and qP were significantly increased under NaCl stress and 10H + N treatment, while Φ_PSII_ and qP were not significantly increased under either NaCl stress or 10H + N treatment (Figure 5C,D).

### 2.5. Effect of Exogenous H_2_O_2_ and NaCl Stress on Antioxidant Enzymes

Compared with CK, CAT, and POD activities of Tartary buckwheat were significantly increased under 5H + N treatment (Figure 6A,B); POD increased nearly 2-fold (189.76%) (Figure 6B), while SOD activity was not significantly increased (Figure 6C), and APX activity was significantly reduced by 47.32% (Figure 6D). Under NaCl stress, CAT, POD, and APX were significantly reduced, and APX decreased the most by 0.09 times what it was under the CK treatment (Figure 6A–D). Compared with NaCl, CAT, POD, SOD, and APX were significantly increased under 5H + N treatment (Figure 6A–D). POD increased the most (11.27-fold) (Figure 6B). POD and APX were significantly increased under 10H + N treatment (Figure 6B,D), while CAT and SOD were not significantly decreased or increased (Figure 6A,C).

Compared with the CK, the GR activity and GSSG content of Tartary buckwheat decreased significantly under NaCl stress, while the AsA content did not increase significantly (Figure 6E–G). GR and AsA were significantly increased under 5H + N treatment, by 70.86% and 54.96%, respectively, while GSSG did not differ significantly from what it was under the CK treatment (Figure 6E–G). AsA and GSSG were significantly decreased under 10H + N treatment, and GR was obviously decreased (Figure 6E–G). Compared with the NaCl stress, GR and GSSG significantly increased under 5H + N treatment, and GR reached a maximum value (0.299) (Figure 6E,G), while the 10H + N treatment significantly increased GR and significantly decreased AsA (Figure 6E,F).

### 2.6. Effect of Exogenous H_2_O_2_ and NaCl Stress on MDA and Phosphoenolpyruvate Carboxylase (PEPC)

The MDA content of Tartary buckwheat reached its maximum value (37.541) under NaCl stress, and its minimum value (17.194) under 5H + N treatment (Figure 7A). Compared with CK, the MDA content was significantly increased under NaCl stress, significantly decreased under 5H + N treatment, and not significantly increased under 10H + N treatment (Figure 7A). Compared with NaCl, the MDA content was significantly decreased by 54.20% and 28.18% under 5H + N and 10H + N treatments, respectively (Figure 7A).

The PEPC activity of Tartary buckwheat in the CK treatment was significantly higher than under NaCl stress and 10H + N treatment (Figure 7B). PEPC decreased most in the 10H + N treatment (59.72%). Under 5H + N treatment, PEPC increased significantly to 1.43 times what it was in the CK treatment (Figure 7B). Compared with NaCl, PEPC was significantly increased by 1.08 times under the 5H + N treatment, and significantly decreased by 40.85% under the 10H + N treatment (Figure 7B).

### 2.7. Effect of Exogenous H_2_O_2_ and NaCl Stress on Relative Expression of Stress Response Genes

As shown in Figure 8A, the relative expression levels of *FtNHX1*, *FtSOS1*, *FtNAC6*, *FtNAC9*, *FtWRKY46*, and *FtbZIP83* were highest in the 5H + N treatment. Relative expression levels of all these genes were greater than 10, with the expression of *FtNHX1* being the highest, followed by that of *FtbZIP83* and *FtNAC9*. The lowest expression of all genes was observed in *FtNAC6* (0.145) under NaCl stress (Figure 8A). Apart from *FtNHX1*, the relative expression levels of other genes were significantly higher in the 5H + N treatment than in the NaCl and 10H + N treatments (Figure 8A). Except for *FtNAC9*, the relative expression levels of other genes were significantly or not significantly lower under NaCl stress than under the 10H + N treatment (Figure 8A).

All genes reached their highest expression level under 5H + N treatment (Figure 8B). *PEPC* showed the highest relative expression (3.533), followed by *MnSOD* (2.962) and *GR* (1.503), and the relative expression levels of genes under 5H + N treatment were significantly higher than they were under NaCl stress and 10H + N treatment (Figure 8B). The relative expression levels of *CuSOD* and *MnSOD* under NaCl treatment were significantly higher (27.21% and 56.10%, respectively) than they were under 10H + N treatment (Figure 8B). For *CAT*, *POD*, and *APX*, the relative expression levels under 10H + N treatment were significantly higher (1.60-fold, 1.99-fold, and 2.07-fold, respectively) than they were under NaCl stress (Figure 8B). However, the relative expression level of *PEPC* did not differ significantly between NaCl stress and 5H + N treatment (Figure 8B).

### 2.8. Effect of Exogenous H_2_O_2_ and NaCl Stress on Relative Expression of Key Genes of the Flavonoid Synthesis Pathway

The relative expression levels of key genes in the flavonoid synthesis pathway were lower under each treatment than under the CK treatment (Figure 8C). The relative expression level of *F3′H* was lowest under NaCl stress (0.059), and highest under the 10H + N treatment (1.047); this was significantly higher than the relative expression levels in the NaCl and 5H + N treatments, by 18.47 times and 4.70 times, respectively. The relative expression levels of *F3H* and *CHS* showed a decreasing trend, and the relative expression levels were highest under NaCl stress (Figure 8C). The *UGT* relative expression level under 5H + N treatment was significantly higher than under the NaCl and 10H + N treatments (Figure 8C).

Cis-acting element analysis (Figure 9) showed that the four key genes in the flavonoid synthesis pathway (*CHS*, *F3′H*, *F3H*, and *UGT*) also contained abscisic-acid-responsive element (abscisic acid responsiveness), methyl-jasmonate-responsive element (MeJA responsiveness), low-temperature-responsive element (low-temperature responsiveness), and other hormones and stress elements, in addition to a large number of light responsiveness elements (light responsiveness). The *F3′H* promoter sequence contained two salt stress response elements (salt stresses) at 986 base pairs (bp) and 1908 bp, respectively (Figure 9). When Tartary buckwheat is subjected to salt stress, its relative expression level of *F3′H* may increase accordingly.

### 2.9. Correlation Analysis, Hierarchical Cluster Analysis, and Principal Component Analysis

Figure 10A shows the results of correlation analysis of exogenous H_2_O_2_ on all growth, photosynthetic pigments, fluorescence parameters, different enzyme activities, and enzyme contents of Tartary buckwheat under NaCl stress. Most of the indicators in the figure are positively correlated, with the highest correlation coefficient between Fv and Fm (r = 0.99), followed by the correlations between Car and Chl b and between Fs and Fv’ (r = 0.97). (Figure 10A). Plant height (PH) was significantly positively correlated with most photosynthetic parameters, such as Fv and Fm (*p* < 0.05). Fm was significantly positively correlated with PH, RF/DW, SDW, and other growth parameters (*p* < 0.05), and PEPC was significantly positively correlated (*p* < 0.05) with leaf indices such as LL and LA (Figure 10A). Figure 10A shows that correlations of AsA and PH, LA, Chl a, and other indicators were not highly significant, and there was no significant negative correlation with Fm’ (*p* > 0.05); SOD, GR, and fluorescence parameters such as Fo, Fv’, Fm’ were significantly lower, and POD showed a low correlation with most enzyme activity/content. In addition, Figure 10A shows that MDA was negatively correlated with all the indicators, and the indicators of enzyme activities and leaf growth, such as GR, CAT, POD, LL, LA, and others, were significantly higher.

In this experiment, principal component analysis was carried out on 28 indices of morphology, physiology, and biochemistry. The two principal components extracted by PCA explained 73.3% and 12.3% of the variation (Figure 10B). The CK and 5H + N treatments showed a positive correlation in PC1 (r > 0), while NaCl and 10H + N were negatively correlated (r < 0). The 10H + N treatments were all on the negative axes of PC1 and PC2, and CK and 5H + N were close together but significantly separated from other treatments. Except for MDA, the other morphological, physiological, and biochemical indicators were positively correlated, which is consistent with the results of the correlation heat map (Figure 10A,B). Photosynthetic indicators such as photosynthetic pigments and fluorescence parameters were on the positive axes of PC1 and PC2, while most parameters, such as enzymes and leaf morphology, were on the positive axis of PC1 and the negative axis of PC2 (Figure 10B). Among the overall scores of the principal components PC1 and PC2, the scores of the CK treatment were closest to those of 5H + N, and the scores of NaCl were closest to those of 10H + N. For both PC1 and PC2, the scores of 5H + N were higher than those of CK.

## 3. Discussion

The impact of soil salinization is becoming increasingly serious on a global scale. About half of the world’s cropland is predicted to be affected by salinization by 2050 [30]. Generally, salinization is due to elevated Na^+^ and K^+^ concentrations in the soil, which result in hyperosmotic conditions that prevent plants from absorbing water, causing adverse effects such as stunted growth [31], osmotic water loss [32], and nutritional imbalance [33]. Salt stress has predominantly inhibitory effects on plant morphogenesis, growth and development, and physiological metabolism. Mitigations and adaptations to the impact of this stress are more obvious in the shoot than in the root system, for example, the reduction in plant height [34], biomass decline [35], and leaf shrinkage [36]. H_2_O_2_ is a relatively stable, freely diffusing, and long-lived reactive oxygen molecule [37], which has a concentration-dependent effect. Exogenous H_2_O_2_ has different regulatory effects in different varieties of plants and different environments. In the present study, the growth of Tartary buckwheat under 5H + N treatment was significantly better than that under the CK, NaCl stress, and 10H + N treatments. Compared with NaCl stress, the PH, SFW/SDW, and RFW/RDW of Tartary buckwheat were significantly increased under 5H + N treatment (*p* < 0.05), particularly the RFW, which increased nearly 1.5-fold (Figure 1 and Figure 2A–E). The above results of this study are consistent with those of Ellouzi et al. [38]. The results of this study indicate that when Tartary buckwheat was subjected to NaCl stress, it is difficult for the roots to absorb water through the conduit to transport it to the stem, leaves, and other tissues of the plant. Appropriate exogenous H_2_O_2_ concentration can alleviate the phenomenon of blocked water absorption under this stress, which is conducive to the growth of the root system and the accumulation of biomass. We also found that the PH, SDW, RFW, and RDW of Tartary buckwheat under 10H + N treatment were slightly higher than they were under NaCl stress, indicating that the treatment had a certain alleviation effect on NaCl stress, but the effect was not obvious. Our results show that in Tartary buckwheat under 5H + N treatment, both LA and LL were significantly increased compared to under CK and NaCl stress treatments, while LRWC was only significantly higher under NaCl stress (Figure 3A–C), which is consistent with Park et al. [39]. In response to salt stress and to protect plants, leaf tissue can reduce the transpiration of leaf water and improve the water retention of leaves by regulating the cuticle and leaf thickness [40]. Therefore, the results of this study show that when the leaf cells were subjected to NaCl stress, the water moved from cells with high water potential to those with low water potential, so that the extracellular water potential was higher than the intracellular water potential. To improve the regulation and adaptability of the plants, the cells reduced transpiration by closing stomata, so that the cells could not absorb further water from the outside.

Chlorophyll is not only an important photosynthetic pigment, but also enables preliminary judgment of leaf function under adverse conditions. Chlorophyll plays an important role in the absorption, transmission, and transformation of light energy, and chlorophyll content is the basis for measuring photosynthetic capacity [41]. Persistent salt stress induces the denaturation of important membrane proteins involved in photosynthesis, enhances the degradation of chlorophyll molecules by chlorophyll-degrading enzymes, and reduces the number of chloroplasts. This is not conducive to the formation and stability of chloroplast ultrastructure, thereby reducing the photosynthetic rate and inhibiting the growth of plants [42,43]. We found that under CK and 5H + N treatments, the Chl a, Chl b, and Car of Tartary buckwheat were significantly higher than under NaCl stress and 10H + N treatments, and values under the CK treatment were not significantly higher than under 5H + N treatment. The change trend of total chlorophyll content (Chl a + b) was the same as that of Chl a and Chl b (Figure 4A–C). This finding is consistent with that of Liu et al. [44]. We found that the Fv/Fm, Fv’/Fm’, and ETR of Tartary buckwheat under 5H + N treatment were significantly increased compared with what they were under NaCl stress, but not significantly different to what they were with the CK treatment (Figure 5A,B,F). This is consistent with the results of He et al. [45]. Sun et al. showed [46] that chlorophyll fluorescence parameters can directly reflect the effects of stress on the activity, function, and electron transfer of PSII in plants. Stress destroys the chlorophyll photosynthetic system and reduces and inhibits the transformation efficiency and potential activity of PSII, resulting in the weakening of plant photosynthetic ability. We also found that the changes in PSII and qP of Tartary buckwheat were opposite to the changes in photosynthetic pigments and other parameters, while NPQ was significantly higher under 5H + N treatment than under NaCl stress (Figure 5C–E). The effect of salt stress on the chlorophyll fluorescence of plants is a complex process. Differences in salt types, plant species, and environmental conditions can affect the main stress mechanism and degree of plants to a large extent, resulting in inconsistent research results. Φ_PSII_ represents the actual photochemical efficiency of PSII under light, and qP reflects the level of photosynthetic activity. Under 150 mmol·L^−1^ NaCl stress, Tartary buckwheat was strongly inhibited. Under 5 or 10 mmol·L^−1^ H_2_O_2_ treatment, electron transfer in the PSII reaction center remained blocked, and the formation of ATP and NADPH assimilation was inhibited [47], while NPQ reflects the ability of plants to dissipate excess light energy into heat [48]. Therefore, the inhibition of photosynthetic fluorescence parameters of Tartary buckwheat under NaCl stress could not be significantly alleviated under H_2_O_2_ treatment. This result is similar to that of Zhang et al. [49].

The salt stress environment affects the normal metabolism of plants, destroys the cell membrane system, generates a large number of ROS, and causes lipid peroxidation of cell membranes. Plants contain antioxidants, including non-enzymatic antioxidants, that can scavenge a large number of ROS under stress conditions, which can alleviate the inhibitory effect of salt stress on plant organisms [50,51]. We found that the activities of CAT, POD, SOD, APX, and GR in Tartary buckwheat were significantly or not significantly higher under 5H + N treatment than under CK or NaCl stress (Figure 6A–E). This indicates that 5 mmol·L^−1^ H_2_O_2_ can significantly enhance the activity of antioxidant enzymes and effectively alleviate the toxic effect of NaCl stress on Tartary buckwheat. This is consistent with the findings of Bhardwaj et al. and Diao et al. [37,52]. We also found that the APX of Tartary buckwheat under 5H + N treatment was significantly lower than under the control treatment (Figure 6D), and the AsA content was significantly lower than that under NaCl stress (Figure 6F). This showed that APX activity and AsA content vary greatly with different plant species and different growth stages. We found that the MDA content of Tartary buckwheat was significantly increased under NaCl stress, but decreased significantly under 5H + N treatment (Figure 7A), which indicated that H_2_O_2_ reduced the formation of ROS in Tartary buckwheat under NaCl stress, attenuated the effect of cell membrane lipid peroxidation, and enhanced the stress resistance of Tartary buckwheat plants. PEPC plays an important role in the balance of carbon and nitrogen metabolism. Studies have shown that PEPC not only plays a key role in the photosynthesis of C4 and CAM plants [53], but also participates in the physiological metabolism of various stress in plants [53], regulates the synthesis and distribution of organic matter in plants, and enhances the carbon skeleton of plants [54]. We found that the PEPC activity of Tartary buckwheat was significantly increased under the 5H + N treatment (Figure 7B), indicating that the PEPC enzyme was involved in the photosynthesis and metabolism of Tartary buckwheat under NaCl stress.

In this study, we found that the relative expression of stress-related genes and related enzyme genes in Tartary buckwheat under 5H + N treatment was significantly increased compared to that under other treatments, and the expression of these genes followed a similar pattern to the morphological indicators and physiological and biochemical changes of Tartary buckwheat. In particular, the expression levels of *FtNHX1*, *FtNAC9*, *FtbZIP83*, *PEPC*, and *MnSOD* were high under 5H + N treatment (Figure 8A,B). These results show that H_2_O_2_ can improve the tolerance of Tartary buckwheat to NaCl stress by regulating the expression of related genes. The expression of *FtNHX1* was significantly increased when treated with 10H + N, which indicates that high concentrations of H_2_O_2_ may also significantly induce the expression of *FtNHX1*. The proteins related to ion transport under salt stress conditions include plasma membrane antiporter SOS1, potassium–sodium co-transporter HKT, and tonoplast antiporter NHX, which can jointly induce Na^+^ accumulation and K^+^ efflux [55,56]. Studies have shown that the expression of Na^+^-induced *SOS1* in Arabidopsis is mediated by H_2_O_2_ induced by NADPH oxidase [57], and SOS3, as a Ca^2+^ sensor, can interact with SOS2 protein kinase. SOS2 compartmentalizes Na^+^/K^+^ in the vacuole by activating vacuolar H^+^-ATPase and NHX, and this compartmentalization plays a crucial role in regulating plant tolerance to salt stress [58]. In this study, the analysis of key genes in the flavonoid synthesis pathway of Tartary buckwheat found that NaCl failed to increase the expression of *CHS*, *F3H*, and *UGT*, while *F3′H* was significantly increased under 10H + N treatment (Figure 8C), indicating that H_2_O_2_ can up-regulate the expression of *F3′H* in Tartary buckwheat under NaCl stress. Through the analysis of *F3H*, *CHS*, *UGT*, and *F3′H* cis-acting elements, we found that *F3′H* contains two salt stress elements (Figure 9), which is consistent with the observed relative expression of *F3′H*.

In this study, we found that the chlorophyll content of Tartary buckwheat was significantly positively correlated with fluorescence parameters (r > 0, *p* < 0.05; Figure 10A). There was a significant positive correlation between Fm and morphological indices (r > 0, *p* < 0.05), a significant positive correlation between PEPC activity and leaf morphology (r > 0, *p* < 0.05), and a significant negative correlation between MDA content and enzyme activity and leaf index (r < 0, *p* < 0.05). PC1 and PC2 identified through principal component analysis explained 73.3% and 12.3% of the variation, respectively (Figure 10B). CK and 5H + N treatments were positively correlated in PC1, and were closest. Except for MDA, the other morphological, physiological, and biochemical indices were clustered in the positive direction PC1, that is, positively correlated, which is consistent with the results of correlation analysis.

This study primarily discussed the regulation of exogenous H_2_O_2_ on the growth, physiology and biochemistry, and related genes of Tartary buckwheat under NaCl stress. Topics for further study include the mechanism by which exogenous H_2_O_2_ regulates the decomposition and distribution of nutrients during the growth of Tartary buckwheat, which metabolic pathways and related enzyme activities and gene expression are mainly regulated, and how to induce and regulate the physiological and biochemical mechanisms of Tartary buckwheat between exogenous H_2_O_2_ and NaCl stress.

## 4. Materials and Methods

### 4.1. Experiment Materials

The tested Tartary buckwheat variety, “Chuanqiao-2”, was provided by the Alpine Crop Research Station (27.96° N, 102.20° E) of Xichang Institute of Agricultural Sciences, Liangshan Prefecture, Sichuan Province, China.

### 4.2. Experiment Design and Treatments

Tartary buckwheat variety “Chuanqiao-2” seeds with uniform size and no pests and diseases were selected. Seeds were rinsed with distilled water, sterilized with 1% NaClO for 10 min, then rinsed with double-distilled water (DDW) several times. The sterilized and rinsed seeds were sown on the surface of 2 layers of qualitative filter paper in a sterile Petri dish (90 mm diameter), and placed in a constant-temperature incubator (25 ± 1 °C during the day, 20 ± 1 °C at night, relative humidity of 75%) for 7 days. At 7 d after sowing (DAS), the Tartary buckwheat seedlings with uniform growth were transplanted into plant pots (diameter 25.5 cm, height =17.5 cm; 3 seedlings per pot) containing mixed nutrient soil (soil:substrate = 1:1) that had been sterilized by high temperature. At 20 d after transplanting (DAT), each pot was irrigated with 2 liters (L) of Hoagland nutrient solution containing 150 mmol·L^−1^ NaCl [59,60]. Based on previous research results [1], 5 or 10 mmol·L^−1^ H_2_O_2_ was used for foliar spraying in this experiment. The transplanted Tartary buckwheat was cultured in a culture room under specific conditions (25 ± 1 °C during the day, 20 ± 1 °C at night, relative humidity of 75%). At 40 days after transplanting (DAT), growth indices and photosynthesis parameters were measured, and sampling was used to measure different physiological and biochemical indices. The schematic diagram of the experimental design is shown in Figure 11. Three replicates were set for each treatment in this experiment, and each treatment was as follows:CK: H_2_O + H_2_O
NaCl: 150 mmol·L^−1^ NaCl + H_2_O
5H + N: 5 mmol·L^−1^ H_2_O_2_ + 150 mmol·L^−1^ NaCl
10H + N: 10 mmol·L^−1^ H_2_O_2_ + 150 mmol·L^−1^ NaCl

### 4.3. Measurement of Plant Height and Biomass

After the plants were uprooted, the surface soil was washed, and the plant height (PH) was measured directly using a ruler, and expressed in cm. The stem fresh weight (SFW) and root fresh weight (RFW) were measured using a thousand-point electronic balance, and then samples were placed in an oven at 105 °C for 30 min, and dried at 65 °C to a constant weight. Stem dry weight (SDW) and root dry weight (RDW) were measured with a thousand-point electronic balance and expressed in mg.

### 4.4. Determination of Leaf Phenotype and Relative Leaf Water Content

Leaf area (LA) and leaf length (LL) were determined with an LI-3000C portable leaf area analyzer (LI-COR, Lincoln, NE, USA), and are expressed in cm^2^ and cm, respectively. Relative water content (LRWC) was determined according to Su et al. [61], using the calculation formula:LRWC (%) = [(FW − DW) × 100]/[TW − DW](1)

### 4.5. Determination of Photosynthetic Pigment Content

The contents of chlorophyll a (Chl a), chlorophyll b (Chl b), and carotenoids (Car) were determined according to Hossain et al. [62]. Briefly, 0.1 g of leaves was cut and placed into 95% (*v*/*v*) ethanol, soaked in dark conditions for 72 h, and then the absorbance at 663 nm, 645 nm, and 470 nm was measured using a microplate reader (Thermo scientific, Waltham, MA, USA). Chl a, Chl b, and Car were determined using the following formulae:Chl a (mg·g^−1^ FW) = [12.7 × (OD663) − 2.69 × (OD645)] × V/(1000 × W)(2)
Chl b (mg·g^−1^ FW) = [22.9 × (OD645) − 4.68 × (OD663)] × V/(1000 × W)(3)
Car (mg·g^−1^ FW) = [100 × (OD470) − 3.27 × Chl a−104 × Chl b]/227(4)

### 4.6. Measurement of Photosynthetic Fluorescence Parameters

The following parameters were measured in the third top-down leaf of the plant’s fully expanded leaves according to the method of Lu et al. [63], using the LI-6400XT portable photosynthesis measurement system (LI-COR, USA): PSII maximum photochemical efficiency (Fv/Fm), minimal fluorescence under dark adaptation (Fo), effective quantum yield (Fv’/Fm’), minimal fluorescence under light (Fo’), stable fluorescence under dark adaptation (Fs), actual photochemical efficiency of PSII (ΦPSII), photochemical quenching coefficient (qP), non-photochemical quenching coefficient (NPQ), and effective electron transfer rate (ETR). The system parameters were set to a light intensity of 1400 μmol·m^−2^·s^−1^, and an atmospheric CO_2_ concentration of 380 ± 5 mmol·mol^−1^. The measurements were conducted between 10 a.m. and 2 p.m. on a sunny day. Parameters were calculated using the following formulae:Fv/Fm = (Fm − Fo)/Fm(5)
Fv’/Fm’ = (Fm’ − Fo’)/Fm’(6)
Φ_PSII_ = (Fm’ − Fs)/Fm’(7)
qP = (Fm’ − Fs)/(Fm’ − Fo’)(8)
NPQ = Fm/Fm’ − 1(9)

### 4.7. Estimation of Biochemical Indicators

Biochemical indicators were estimated using the BOXBIO kit (Beijing Boxbio Science & Technology Co., Ltd., Beijing, China) following the manufacturer instructions.

#### 4.7.1. Antioxidant Enzyme Activity

Catalase activity (CAT), peroxidase activity (POD), superoxide dismutase activity (SOD), ascorbate peroxidase activity (APX), and glutathione reductase activity (GR) were estimated according to Kong et al. [64]. The corresponding extract was added to 0.1 g of sample, followed by 1 mL of extract, and the supernatant was collected. Identical tubes that were not illuminated served as blanks. Absorbance values for CAT, POD, SOD, APX, and GR were measured at 240 nm, 470 nm, 560 nm, 290 nm, and 340 nm, respectively, using a spectrophotometer (Unico Instrument Co., Ltd., Shanghai, China) and expressed in U·g^−1^.

#### 4.7.2. Antioxidant Enzyme Content

Ascorbic acid content (AsA) and oxidized glutathione content (GSSG) were measured according to Rama and Prasad [65]. Homogenization was conducted in an ice bath: the sample was centrifuged at 8000× *g* at 4 °C to collect the supernatant and then placed on ice for measurement. A microplate reader (Thermo scientific, Waltham, MA, USA) was used at 265 nm and 412 nm, respectively, after adding related reagents. The absorbance values for AsA and GSSG are expressed in nmol·g^−1^ and ug·g^−1^, respectively.

#### 4.7.3. Malondialdehyde Content

Malondialdehyde content was measured according to Hodges et al. [66] We took 0.1 g of sample and homogenized the supernatant for subsequent detection. The MDA content was calculated according to the corresponding absorbance and formula. The absorbance was measured at 450 nm, 532 nm, and 600 nm with a visible spectrophotometer (Unico Instrument Co., Ltd., Shanghai, China), and expressed in nmol·g^−1^.

#### 4.7.4. Phosphoenolpyruvate Carboxylase (PEPC) Activity

PEPC activity was measured according to Aoyagi and Bassham [67]. The samples were processed according to the ratio of sampling mass and extract volume to 1:10, and the corresponding reagents were added for measurement after collecting the supernatant. Absorbance at 340 nm was measured with an ultraviolet spectrophotometer (Unico Instrument Co., Ltd., Shanghai, China) and expressed in U·g^−1^.

### 4.8. Total RNA Extraction and Reverse Transcription PCR (RT-PCR)

A 0.1 g sample of Tartary buckwheat leaves was fully ground in liquid nitrogen, and then total RNA was extracted using the E.Z.N.A. Plant RNA Kit (Omega Bio-tek, Inc., Norcross, GA, USA). The integrity of extracted RNA was checked by 1% agarose gel electrophoresis, and RNA purity and concentration were measured using a spectrophotometer (Beijing Kaiao Technology Development Co., Ltd., Beijing, China). RNA was reverse-transcribed into the first-strand cDNA using the Hiscript II Q RT Supermix for qPCR Kit (Vazyme Biotech Co., Ltd., Nanjing, China) in a 20 μL reaction system.

### 4.9. Quantitative Real-Time PCR (qRT-PCR) Analysis

Primer Premier 5.0 (Premier Corporation, Vancouver, BC, Canada) software was used to design specific primers for qPCR, with *FtH3* as the internal control gene, and the amplification primers are shown in Table 1. The ChamQ Universal SYBR qPCR Master Mix Kit (Vazyme Biotech Co., Ltd., Nanjing, China) was used with 1.0 μL of template cDNA, 10.0 μL of 2× SYBR Mix, and 0.4 μL each of Primer F and R, and the reaction volume was made up to 20 μL with ddH_2_O. qPCR was performed using CFX96 Real-Time System (BIO-RAD, Hercules, CA, USA). The relative expression levels of target genes compared to the internal control gene were calculated using the 2^−ΔΔCt^ method [68].

### 4.10. Prediction of Cis-Acting Element

The PlantCare website (We accessed on 14 July 2021 to http://bioinformatics.psb.ugent.be/webtools/plantcare/html/) was used to predict the presence of cis-acting elements in promoter sequences (the upstream 2000 bp) of key genes of the flavonoid synthesis pathway.

### 4.11. Statistics and Analysis

Analysis of variance (ANOVA, *p* < 0.05), multiple comparisons (Duncan), and correlation analysis (Pearson) were performed with IBM SPSS Statistics 26.0 software (International Business Machines Corporation, New York, NY, USA), and the results are expressed as the mean ± SD. Column charts were drawn using GraphPad Prism7.0 software (GraphPad Software, LLC, San Diego, CA, USA). Principal component analysis plots were made using Origin 2019b software. Correlation heatmaps and cis-acting element visualizations were generated using TBtools v1.09876 [69].

## 5. Conclusions

In this study, compared with CK and NaCl treatments, spraying exogenous H_2_O_2_ could promote the growth of Tartary buckwheat under NaCl stress, increase the accumulation of chlorophyll content, enhance electron transfer and transformation during photosynthesis, effectively improve enzymatic reactions, reduce cell membrane lipid peroxidation, induce or activate the expression level of related genes, alleviate the toxic effect of NaCl stress on Tartary buckwheat, and promote normal physiological metabolism and biochemical reactions in Tartary buckwheat. A concentration of 5 mmol·L^−1^ H_2_O_2_ produced the optimal promoting effect on Tartary buckwheat under 150 mmol·L^−1^ NaCl stress. Appropriate concentrations of H_2_O_2_ can alleviate the inhibitory effect of salt stress, but the mechanisms and signaling pathways of H_2_O_2_-mediated salt tolerance still need to be further dissected in detail, as well as how H_2_O_2_ regulates related genes to activate defense systems to alleviate salt stress.

## Figures and Tables

**Figure 1 ijms-23-10698-f001:**
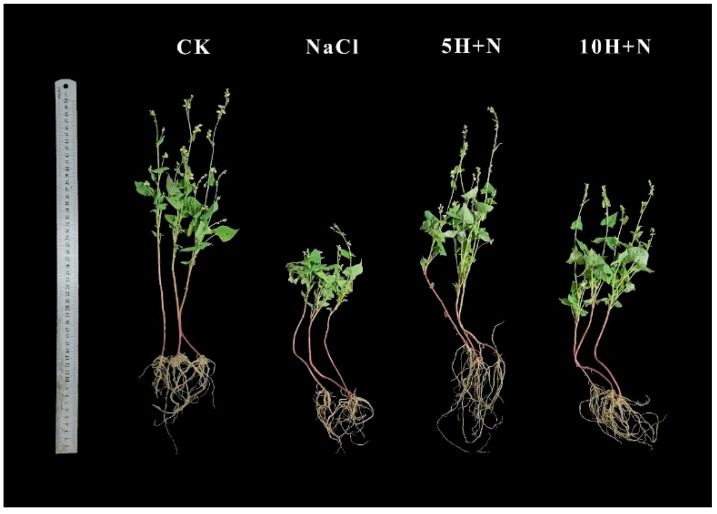
Effects of exogenous H_2_O_2_ on growth of Tartary buckwheat under NaCl stress. The photos were taken on the 40th day of transplanting. CK: control, H_2_O foliar spray + H_2_O irrigation, NaCl: H_2_O foliar spray + 150 mmol·L^−1^ NaCl irrigation, 5H + N: 5 mmol·L^−1^ H_2_O_2_ foliar spray + 150 mmol·L^−1^ NaCl irrigation, 10H + N: 10 mmol·L^−1^ H_2_O_2_ foliar spray + 150 mmol·L^−1^ NaCl irrigation.

**Figure 2 ijms-23-10698-f002:**
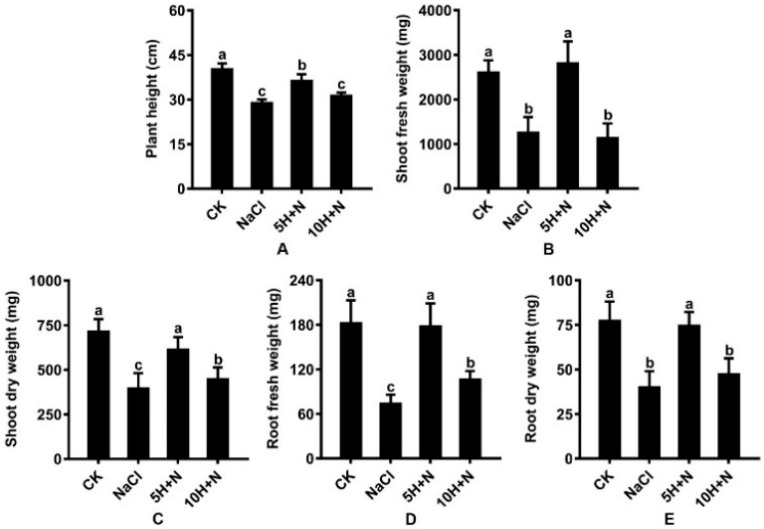
Effects of exogenous H_2_O_2_ on PH (**A**), SFW (**B**), SDW (**C**), RFW (**D**), and RDW (**E**) of Tartary buckwheat under NaCl stress. CK: control, H_2_O foliar spray + H_2_O irrigation, NaCl: H_2_O foliar spray + 150 mmol·L^−1^ NaCl irrigation, 5H + N: 5 mmol·L^−1^ H_2_O_2_ foliar spray + 150 mmol·L^−1^ NaCl irrigation, 10H + N: 10 mmol·L^−1^ H_2_O_2_ foliar spray + 150 mmol·L^−1^ NaCl irrigation. PH: plant height (cm), SFW: shoot fresh weight (mg), SDW: shoot dry weight (mg), RFW: root fresh weight (mg), RDW: root dry weight (mg). All values are expressed as mean ± SD. According to Duncan’s multiple comparisons, different letters represent significant differences among different treatments (*p* < 0.05).

**Figure 3 ijms-23-10698-f003:**
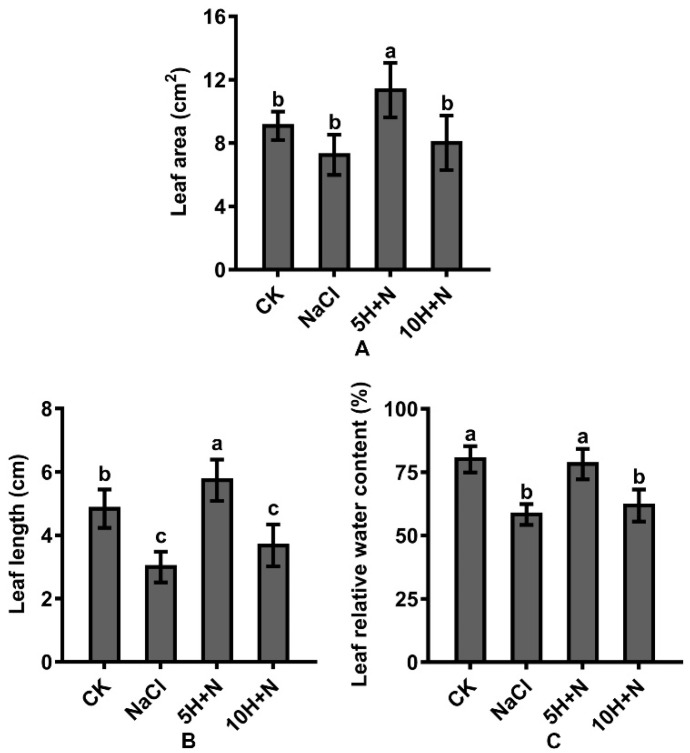
Effects of exogenous H_2_O_2_ on LA (**A**), LL (**B**), and LRWC (**C**) of Tartary buckwheat under NaCl stress. CK: control, H_2_O foliar spray + H_2_O irrigation, NaCl: H_2_O foliar spray + 150 mmol·L^−1^ NaCl irrigation, 5H + N: 5 mmol·L^−1^ H_2_O_2_ foliar spray + 150 mmol·L^−1^ NaCl irrigation, 10H + N: 10 mmol·L^−1^ H_2_O_2_ foliar spray + 150 mmol·L^−1^ NaCl irrigation. LA: leaf area (cm^2^), LL: leaf length (cm), LRWC: leaf relative water content (%). All values are expressed as mean ± SD. According to Duncan’s multiple comparisons, different letters represent significant differences among different treatments (*p* < 0.05).

**Figure 4 ijms-23-10698-f004:**
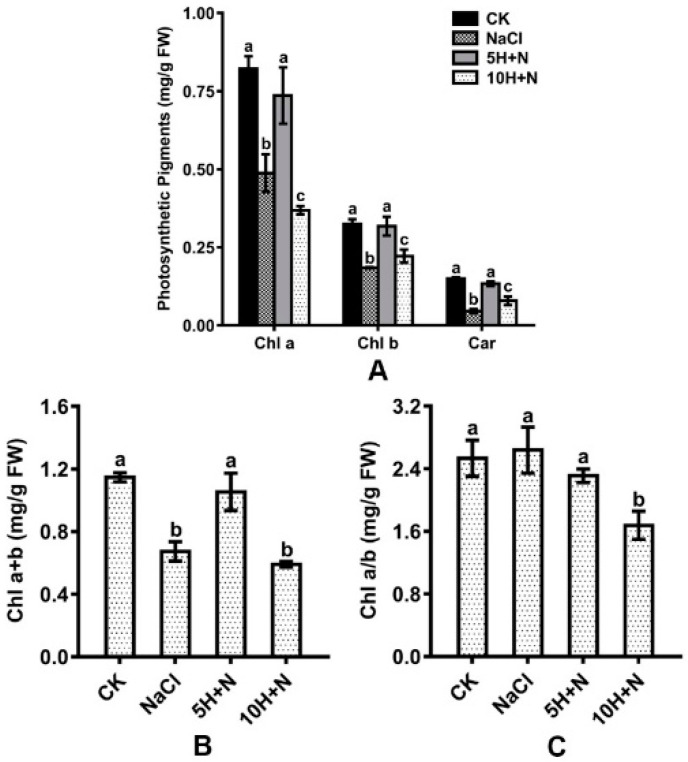
Effects of exogenous H_2_O_2_ on Chl a, Chl b, and Car (**A**), Chl a + b (**B**), and Chl a/b (**C**) of Tartary buckwheat under NaCl stress. CK: control, H_2_O foliar spray + H_2_O irrigation, NaCl: H_2_O foliar spray + 150 mmol·L^−1^ NaCl irrigation, 5H + N: 5 mmol·L^−1^ H_2_O_2_ foliar spray + 150 mmol·L^−1^ NaCl irrigation, 10H + N: 10 mmol·L^−1^ H_2_O_2_ foliar spray + 150 mmol·L^−1^ NaCl irrigation. Chl a: chlorophyll a (mg/g FW), chlorophyll b (mg/g FW), Car (mg/g FW). All values are expressed as mean ± SD. According to Duncan’s multiple comparisons, different letters represent significant differences among different treatments (*p* < 0.05).

**Figure 5 ijms-23-10698-f005:**
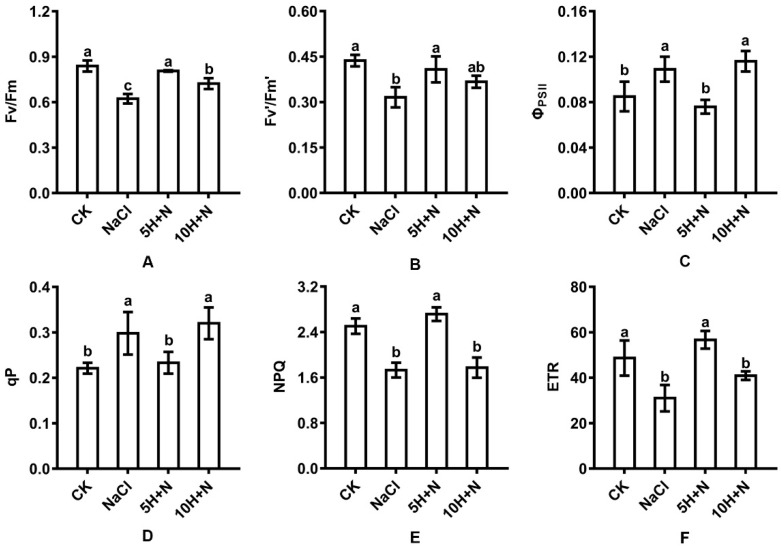
Effects of exogenous H_2_O_2_ on Fv/Fm (**A**), Fv’/Fm’ (**B**), Φ_PSII_ (**C**), qP (**D**), NPQ (**E**), and ETR (**F**) of Tartary buckwheat under NaCl stress. CK: control, H_2_O foliar spray + H_2_O irrigation, NaCl: H_2_O foliar spray + 150 mmol·L^−1^ NaCl irrigation, 5H + N: 5 mmol·L^−1^ H_2_O_2_ foliar spray + 150 mmol·L^−1^ NaCl irrigation, 10H + N: 10 mmol·L^−1^ H_2_O_2_ foliar spray + 150 mmol·L^−1^ NaCl irrigation. Fv/Fm: PSII maximum photochemical efficiency, Fv’/Fm’: effective quantum yield, Φ_PSII_: actual photochemical efficiency of PSII, qP: photochemical quenching coefficient, NPQ: non-photochemical quenching coefficient, ETR: effective electron transfer rate. All values are expressed as mean ± SD. According to Duncan’s multiple comparisons, different letters represent significant differences among different treatments (*p* < 0.05).

**Figure 6 ijms-23-10698-f006:**
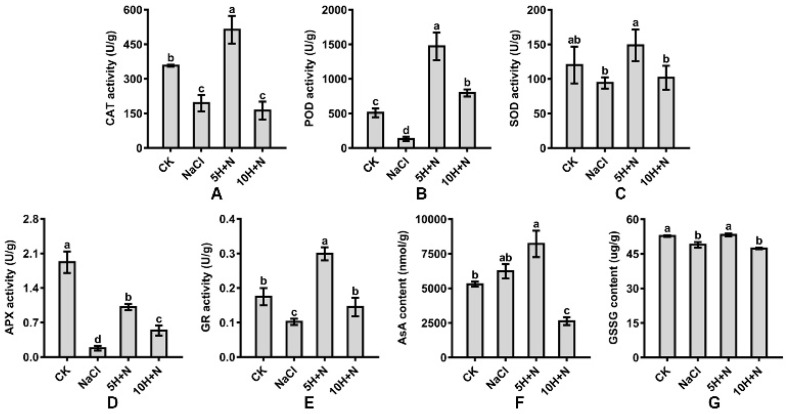
Effects of exogenous H_2_O_2_ on CAT (**A**), POD (**B**), SOD (**C**), APX (**D**), GR (**E**), AsA (**F**), and GSSG (**G**) of Tartary buckwheat under NaCl stress. CK: control, H_2_O foliar spray + H_2_O irrigation, NaCl: H_2_O foliar spray + 150 mmol·L^−1^ NaCl irrigation, 5H + N: 5 mmol·L^−1^ H_2_O_2_ foliar spray + 150 mmol·L^−1^ NaCl irrigation, 10H + N: 10 mmol·L^−1^ H_2_O_2_ foliar spray + 150 mmol·L^−1^ NaCl irrigation. CAT: catalase activity, POD: peroxidase activity, SOD: superoxide dismutase activity, APX: ascorbate peroxidase activity, GR: glutathione reductase activity, AsA: ascorbic acid content, GSSG: glutathione oxidized content. All values are expressed as mean ± SD. According to Duncan’s multiple comparisons, different letters represent significant differences among different treatments (*p* < 0.05).

**Figure 7 ijms-23-10698-f007:**
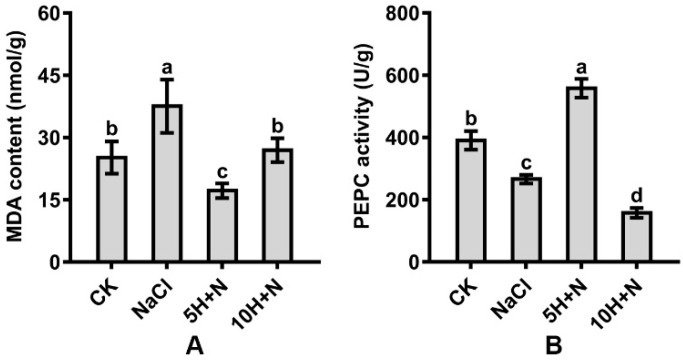
Effects of exogenous H_2_O_2_ on MDA content (**A**) and PEPC activity (**B**) of Tartary buckwheat under NaCl stress. CK: control, H_2_O foliar spray + H_2_O irrigation, NaCl: H_2_O foliar spray + 150 mmol·L^−1^ NaCl irrigation, 5H + N: 5 mmol·L^−1^ H_2_O_2_ foliar spray + 150 mmol·L^−1^ NaCl irrigation, 10H + N: 10 mmol·L^−1^ H_2_O_2_ foliar spray + 150 mmol·L^−1^ NaCl irrigation. MDA: malondialdehyde content, PEPC: phosphoenolpyruvate carboxylase activity. All values are expressed as mean ± SD. According to Duncan’s multiple comparisons, different letters represent significant differences among different treatments (*p* < 0.05).

**Figure 8 ijms-23-10698-f008:**
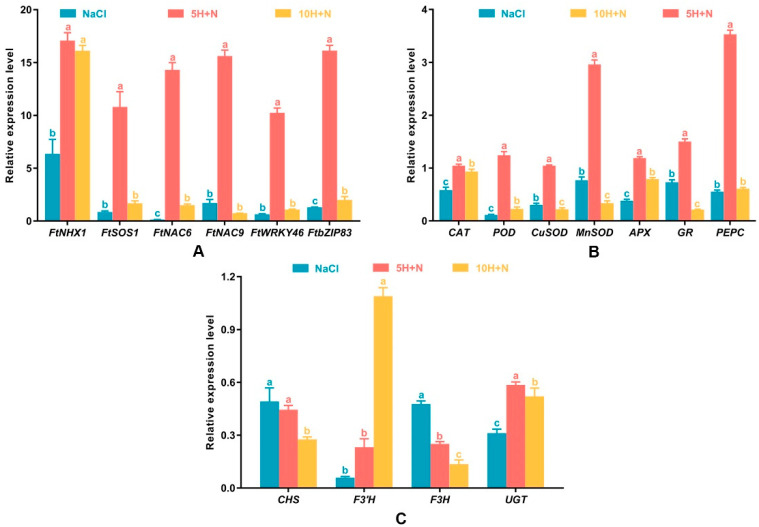
Effects of exogenous H_2_O_2_ on the relative expression levels of stress-related genes (*FtNHX1*, *FtSOS1*, *FtNAC6*, *FtNAC9*, *FtWRKY46*, and *FtbZIP8*) (**A**), enzyme-related genes (*CAT*, *POD*, *CuSOD*, *MnSOD*, *APX*, *GR*, and *PEPC*) (**B**), key genes (*CHS*, *F3′H*, *F3H*, and *UGT*) in the flavonoid synthesis pathway (**C**) of Tartary buckwheat under NaCl stress. NaCl: H_2_O foliar spray + 150 mmol·L^−1^ NaCl irrigation, 5H + N: 5 mmol·L^−1^ H_2_O_2_ foliar spray + 150 mmol·L^−1^ NaCl irrigation, 10H + N: 10 mmol·L^−1^ H_2_O_2_ foliar spray + 150 mmol·L^−1^ NaCl irrigation. All values are expressed as mean ± SD. According to Duncan’s multiple comparisons, different letters represent significant differences among different treatments (*p* < 0.05).

**Figure 9 ijms-23-10698-f009:**
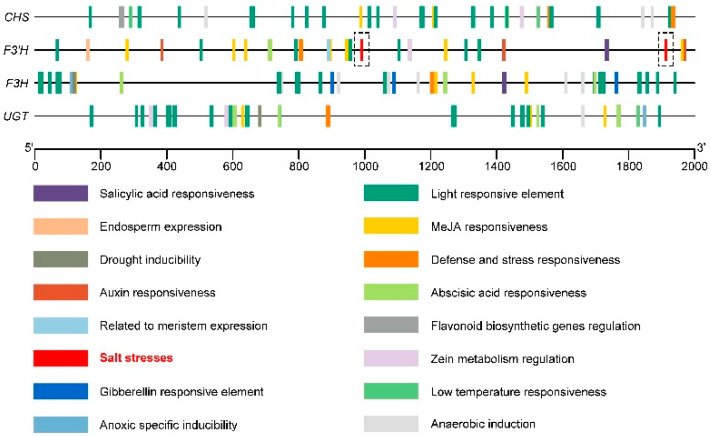
The cis-acting element of the promoter sequence (upstream 2000 bp) of the key genes *CHS*, *F3′H*, *F3H*, and *UGT* in the Tartary buckwheat flavonoid synthesis pathway.

**Figure 10 ijms-23-10698-f010:**
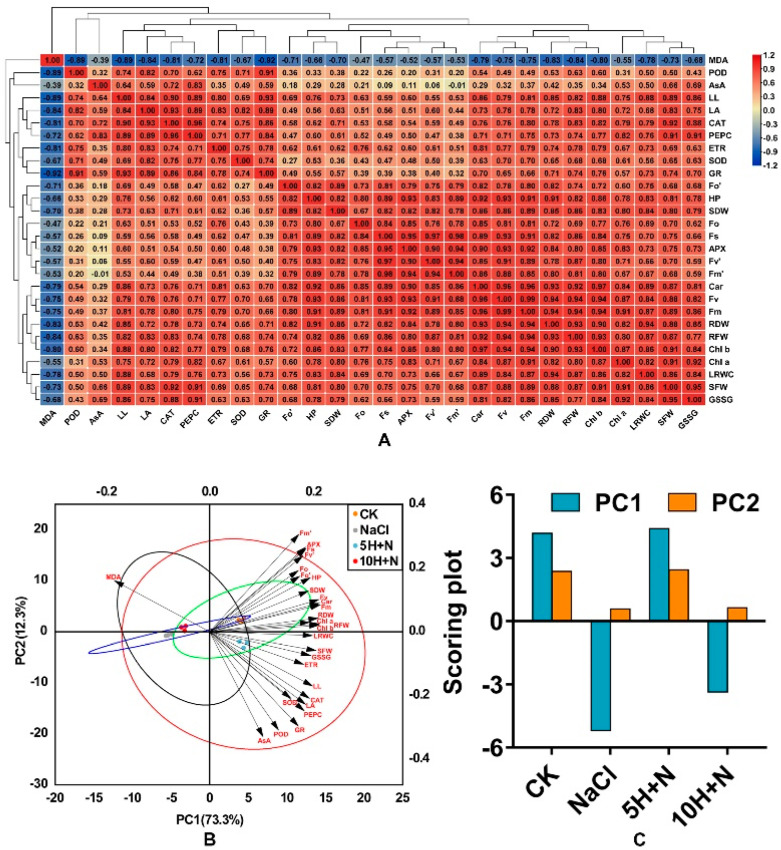
Correlation hierarchical cluster analysis of all indices. Positive number: positive correla−tion; negative number: negative correlation. Red frames indicate significant positive correlation (*p* < 0.05); blue frames indicate significant negative correlation (*p* > 0.05) (**A**). Principal component analysis (PCA) among all indices and treatments (**B**) and scoring plot of principal component anal−ysis among all treatments. (**C**) CK: control, H_2_O foliar spray + H_2_O irrigation, NaCl: H_2_O foliar spray + 150 mmol·L^−1^ NaCl irrigation, 5H + N: 5 mmol·L^−1^ H_2_O_2_ foliar spray + 150 mmol·L^−1^ NaCl irrigation, 10H + N: 10 mmol·L^−1^ H_2_O_2_ foliar spray + 150 mmol·L^−1^ NaCl irrigation.

**Figure 11 ijms-23-10698-f011:**
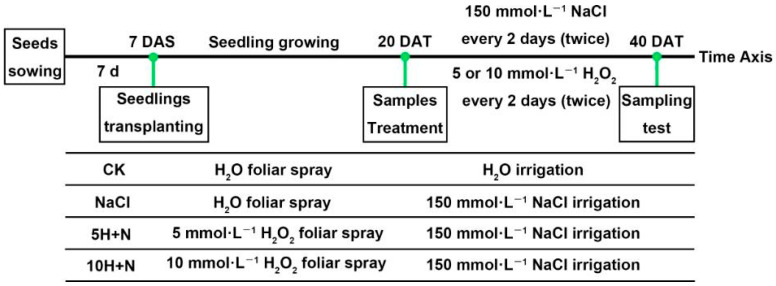
Schematic diagram of experimental design. 7 DAS: 7 days after sowing, 20 DAT: 20 days after transplanting. CK: control, H_2_O foliar spray + H_2_O irrigation, NaCl: H_2_O foliar spray + 150 mmol·L^−1^ NaCl irrigation, 5H + N: 5 mmol·L^−1^ H_2_O_2_ foliar spray + 150 mmol·L^−1^ NaCl irrigation, 10H + N: 10 mmol·L^−1^ H_2_O_2_ foliar spray + 150 mmol·L^−1^ NaCl irrigation.

**Table 1 ijms-23-10698-t001:** The primer sequences for qPCR.

Primer Name	Forward Primer (5′-3′)	Reverse Primer (5′-3′)	Functions
*FtH3*	AATTCGCAAGTACCAGAAG	CCAACAAGGTATGCCTCAGC	Actin
*FtNHX1*	CGTTGCTAGGACGCAATGTTCCA	ACAGTCCACGTCGGATGCCTTAT	Stress-related genes
*FtSOS1*	CCTTACACCGTACCCGCTC	CCGGAAGAAACACAGCCAACA	
*FtNAC6*	GATTCAATTCCCCGGCTCCA	AACGGGGACAACTCATTCCC	
*FtNAC9*	CTGAGGGTGTAATTCCGGGT	TCAACGGTAGGGGTAGAAGC	
*FtWRKY46*	TGTTCCGCCTTCTGATGGTT	CAGCACTGTGGGGTCATCAT	
*FtbZIP83*	ACCGAGTATTCCGCAAGCTC	AACTCTCCCCAAAACCCACC	
*CAT*	GAGTTTGGTTCCCTTGCTT	TTCATACACTTCACTGGCGT	Enzyme-related genes
*POD*	GTTCTGGTTGGGCTTGG	TTGTCCTCGTCTGTTGGTC	
*CuSOD*	ATGGTGCTCCTGACGATG	CCACTGCCCTTCCAATAAT	
*MnSOD*	GTCTACGGTCCTCCTTCTACAT	TAACAACAGCACACTTCTTTCT	
*APX*	TACTCCGAGGTTGTGTGCC	CAATCAAGGTGTTCCAGTCA	
*GR*	TTGAGTGGAGAGAAGGAAGG	CATAGTCGGCAAAGAAAGC	
*PEPC*	AAGTCTCCACATTCGGTCTC	ATCTCCAAGTGCCTGGTTAT	
*CHS*	GAGGAGATCAGAAAGGCACAAAGGG	GTCGGCTTGGTAGATACAGTTAGGC	Flavonoid-synthesis-related genes
*F3′H*	TCAAGGAGAATGGCGGAGTT	TGGGTGGTTCAGGAGGAGTG	
*F3H*	GCCTGTTGAGGGTGCCTTTGTC	TGGCGATTGAAAGACGGCTGAAG	
*UGT*	CAGCTTCTTCACCACCGAATCCTC	TCTCGCCCGCTAACCCATCTTC	

## Data Availability

Data are contained within the article.

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
