# Peer review of "Physiological and Biochemical Regulation Mechanism of Exogenous Hydrogen Peroxide in Alleviating NaCl Stress Toxicity in Tartary Buckwheat (Fagopyrum tataricum (L.) Gaertn)"

_ijms, 2022, doi:10.3390/ijms231810698_

Round 1

Reviewer 1 Report

Manuscript: Physiological and biochemical regulation mechanism of exog-enous
hydrogen peroxide in alleviating NaCl stress toxicity in tartary buckwheat
(Fagopyrum tataricum (L.) Gaertn)
Journal: International Journal of Molecular Sciences

The manuscript reports experimental results on the biochemical and physiological mechanism  induced by hydrogen peroxide to alleviate salt stress toxicity in tartary buckwheat.

- Paper important and interesting.

- I have only a formal revision about the caption of almost all figures, which are hard to read in distinguishing the samples.

- The references are OK.

Make more clear the figures,. sometimes the labels are really hard to read.

- Please, Revise the English.

Minor revision is demanded.

Author Response

- I have only a formal revision about the caption of almost all figures, which are hard to read in distinguishing the samples.

The titles of all figures in the manuscript have been revised.

Make clearer the figures, sometimes the labels are really hard to read.

We've made almost all clearer figures.

- Please, Revise the English.

We have asked native English-speaking editors to retouch the English of the manuscript.

Reviewer 2 Report

Dear authors,

I get your article (Physiological and biochemical regulation mechanism of exog-enous hydrogen peroxide in alleviating NaCl stress toxicity in tartary buckwheat (Fagopyrum tataricum (L.) Gaertn) to review. Your article is very well organized and written as well as have parameters tested and your article.  However, the using of H2O2 as foliar spray to mitigate salinity was studied in several crops many times. But, the article could be publishing because you work in new crop. Also, please revised my comments in attached pdf file and reply to every comment and mention the lines.

All the best

Author Response

We have revised the manuscript based on your revision comments and have responded to each comment with a reference to the lines.

Line 16 of the abstraction section has been modified.

Line 43 of the introduction section has been modified.

According to your suggestion, I have added relevant content in the introduction section.

Line 100 of the results section has been modified.

Line 201, 203, 204, 211,213, 310, 312, 314, 358 and 366 of the results section has been modified.

Line 502, 515 537, 546, 562, 571, 572, 575 and 581 of the materials and methods section has been modified.

Line 617, 624-628 of the conclusion section has been modified.

Round 2

Reviewer 2 Report

Thank you for your efforts in improve the article quality. The article is suitable now for publish in the curent form.